# D2D Communication Network Interference Coordination Scheme Based on Improved Stackelberg

**Xinzhou Li, Guifen Chen \*, Guowei Wu, Zhiyao Sun and Guangjiao Chen**

School of Electronic and Information Engineering, Changchun University of Science and Technology, Changchun 130022, China
\* Correspondence: 2019100482@mails.cust.edu.cn

**Abstract:** The sudden explosive growth of data in intelligent devices and existing communication networks has brought great challenges to existing communication networks. On the one hand, D2D (device to device) technology greatly improves the utilization of spectrum resources; on the other hand, it improves the communication quality of users. It has become an important part of the future communication network. Aiming at the problem that the existing D2D communication network system has complex user interference, and the communication quality of cellular users is difficult to guarantee, a D2D communication network interference coordination scheme based on improved Stackelberg is proposed. Using resource allocation and power control to solve the interference co-ordination problem, this paper proposes an improved Stackelberg model based on DQN (deep Q network), establishes the master–slave game between cellular users and multiplexing resource users (D2D users; relay communication users), optimizes the cost parameters in the Stackelberg mode and improves the transmission power and resource allocation scheme of multiplexing resource users. The simulation results show that compared with similar algorithms, the algorithm proposed in this paper has the best performance in guaranteeing the QoS of cellular users in the system and has good interference management capability for D2D communication networks.

**Keywords:** D2D communication; deep reinforcement learning; Stackelberg game; relay communication; interference coordination

## 1. Introduction

The commercialization of 5G has greatly promoted the construction of smart cities, the development of smart transportation and the realization of the idea of the interconnection of everything [1,2]. The new generation of application scenarios such as We Media era, new video conference, mobile teaching, AR, VR, meta universe, etc., also makes the number of intelligent terminals show an explosive growth trend [3,4]. According to the statistics, it is estimated that, in 2040, the number of intelligent terminal connections will increase by more than 30 times compared with 2022, and the average monthly traffic will increase by more than 130 times [5]. Finally, in the 6G era, there will be an Internet market of "hundreds of billions of terminal connections, trillions of gigabytes of average monthly traffic" [6]. At the same time, the limited spectrum is constantly being developed and utilized, and the shortage of spectrum resources is becoming increasingly serious. How to ensure people's growing demand for communication traffic and rationally solve the problem of spectrum resource shortage are the main issues that people need to deal with in today's society. Fortunately, the arrival of relay communication technology and device to device (D2D) communication technology has brought solutions to the above problems [7,8].

Relay communication technology and D2D communication technology are added to the cellular edge cells [9]. On the one hand, relay communication can realize the normal

communication of the edge cellular users through two or more hops, greatly improving their own user communication experience and reducing the system energy consumption [10]. On the other hand, the addition of D2D communication strengthens the whole wireless communication system and reduces the computing pressure of the base station to a certain extent, while improving the spectrum resource utilization and throughput of the system [11].

The integration of relay communication technology, D2D communication technology and cellular communication is the inevitable trend of future communication. However, the converged heterogeneous cellular network will produce unavoidable interference, which includes the interference generated by multiple users using different communication methods and the same layer and inter layer interference generated by D2D users when reusing cellular user communication [12,13]. Therefore, it is of great significance to design a reasonable and effective interference coordination scheme to improve the system throughput while ensuring the user communication quality.

In recent years, many domestic and foreign research institutions, experts and scholars have carried out a lot of research on the interference problems in the D2D network. The related research work can be generally divided into starting from relay-aided communication and D2D user multiplexing cellular user communication, as well as considering the above two communication methods to achieve the interference management of the D2D communication network.

In the relay-assisted D2D communication network, the literature [14] first senses the surrounding D2D devices by relaying, establishes a priority table, then uses the Gale–Shapley algorithm to match resources stably and reliably, and finally establishes an incentive mechanism to weaken interference. The proposed scheme improves the system throughput and user and average satisfaction before ensuring low interference. The literature [15] proposed a D2D relay anti-interference algorithm based on reverse auction. Compared with the auction based on the Vickrey–Clark–Groves (VCG) mechanism, the proposed algorithm has obvious advantages in terms of user data rate, all interference of the base station, system throughput, etc. The literature [16] proposed a relay communication selection mechanism based on optimal weight. The proposed algorithm ensures the average user experience of the system and has relatively low outage probability. The literature [17] proposed a relay-assisted D2D transmission scheme based on energy collection. The relay with energy collection is placed in the cellular network to establish user information transmission constraints. Finally, under the premise of ensuring the communication quality of cellular users, the sum rate of D2D users is greatly optimized. The literature [18] proposed a bidirectional relay D2D communication model. Each relay in the model assists a D2D communication link. On the one hand, it collects energy, and on the other hand, it redistributes power, which improves the energy efficiency of the system and reduces system interference. The literature [19] proposed a relay-assisted D2D communication algorithm based on the depth deterministic policy gradient (DDPG). The algorithm improves the transmission rate of D2D communication users and the energy efficiency of the system through the uninterrupted information interaction parameters in the network. According to the selfishness attribute of users in the relay communication network and the limited willingness attribute of transmitting relay data, the literature [20] proposes a potential relay selection identification scheme, which improves the incentive of each user in the system and improves the communication quality of users and the system throughput.

In the communication network of D2D multiplexing cellular users, the literature [21] proposed a D2D user power algorithm in the bidirectional relay scenario, which maximizes the system throughput while ensuring the SINR of users in the system. According to the transmission power budget (interference coordination cost) of different users, the literature [22] proposes a joint power control scheme of source and relay nodes based on deep learning, which improves the sum rate of the network. The literature [23] established a Stackelberg game model of the master–slave game of the base station and D2D users by

using game theory and used the probability threshold method to constrain relevant parameters, which effectively alleviated system interference and improved the user energy efficiency of the system. The literature [24] proposed a D2D communication bargaining game scheme to replace the quotation. In the scheme, D2D users are compared to cell users, and the transmission power is used as reward and punishment to improve the total rate of the system while ensuring the SINR of users in the system. The literature [25] proposed a D2D heterogeneous network interference scheme based on alliance games, in which the transmission rate of users in the D2D system is improved at the cost of the residual self-interference (RSI) of users.

Finally, an interference coordination scheme is considered from multiple perspectives. The paper [26] considers both relay communication and D2D user communication and proposes a centralized hierarchical interference management algorithm based on deep reinforcement learning. The proposed algorithm is close to the centralized D2D communication interference coordination algorithm with full information interaction in terms of convergence and time delay. The literature [27] proposed a joint optimization communication interference coordination scheme. The proposed scheme simultaneously selects the link transmission rate and user output power, reducing interference and energy consumption in the system. The reference [28] proposed a joint optimization algorithm using graph coloring. The proposed algorithm jointly optimizes the communication mode and resource allocation of D2D users, as well as the relay selection, to ensure the SINR of cellular and D2D users in the system. The literature [29] also considers relay selection, user association and user resource allocation, establishes a new relay communication framework, effectively coordinates interference in the system and improves the average transmission rate of users. The literature [30] uses a Q-learning algorithm to optimize the relay selection problem in D2D communication systems and uses the Dinkelbach method and Lagrange dual decomposition method to solve the power allocation problem of users. The proposed scheme reasonably and effectively handles the interference in the system and maximizes the total energy efficiency of D2D users. The literature [31] proposed a joint interference coordination framework based on distance, SINR, transmission power and user QoS, which improved the user access rate of the network and the sum rate of the system.

From the above development and research of interference coordination algorithms, it can be found that it has become a trend to consider the D2D network interference management scheme from multiple perspectives, but there is relatively little research on the user communication experience. Based on this, this paper starts from the transmission rates of both relay and D2D users, adds the idea of dynamic game and carries out research on the D2D communication interference coordination scheme based on repeated games to improve the transmission rate of users. See Table 1 for the relevant information.

**Table 1.** Related works.

| Reference | Considered Angles | Optimization Idea | Optimization Direction |
|---|---|---|---|
| [14,17,20] | Relay | Relay sensing | System throughput; energy efficiency |
| [15,18] | Relay | Auction idea | Data rate; energy efficiency |
| [16,19] | Relay | Iteration idea | Outage probability; energy efficiency |
| [21,22] | D2D | Power control | System throughput; data rate |
| [23–25] | D2D | Game theory | Data rate; energy efficiency |
| [26,30] | Relay + D2D | Auction idea | Astringency; time delay; QoS |
| [27,29,31] | Relay + D2D | Joint optimization | System throughput; access rate |
| [28] | Relay + D2D | Graph theory | System throughput; data rate |

The major contributions of the proposed work are furnished as follows:
(1) We propose a DQN–Stackelberg model framework, in which the base station is the leader, and both D2D and relay users are the followers.

(2)    An interference management algorithm for heterogeneous D2D cellular communication networks is proposed, and a power control and channel allocation scheme is designed to solve the interference problem in D2D communication networks.

(3)    The system simulation experiment first proves the effectiveness of the proposed algorithm. At the same time, the comparison of related algorithms shows that the proposed algorithm effectively guarantees the communication quality of cellular users and improves the system throughput and interference management capability.

The structure of the paper is as follows: The second section introduces the model of the D2D communication system. The third section introduces the interference management algorithm proposed in this paper, in detail; the fourth section carries out the simulation verification of the algorithm proposed in this paper, as well as the comparative experiment and experimental analysis of related algorithms; and the fifth section carries out the summary of the full text and future work plan.

## 2. System Model

This paper considers the scenario of single base station D2D users reusing cellular users' uplink resources. The system model includes D2D user pairs, relay users, cellular users, relay nodes and base stations. There are many kinds of interference in the system. When D2D users reuse the uplink communication resources of cellular users, the base station is subject to signal interference from the transmitter of D2D users. During normal communication between the cellular user and the BTS, the D2D user receiver receives signal interference from the cellular user transmitter. When relay users reuse cellular uplink communication resources, the base station is subject to signal interference from the transmitting end of the relay users, and the receiving end of the relay users is subject to signal interference from the transmitting end of the cellular users.

### 2.1. Establishment of System Model

Shown in Figure 1 is a schematic diagram of the transmission of many to one multiplexed uplink communication system, taking a group of D2D, cellular and relay users as examples. The system model has a total of one base station, one relay node, M cellular users (j = 1, 2, 3, ..., M), N pairs of D2D users (i = 1, 2, 3, ..., N) and Q relay users (q = 1, 2, 3, ..., Q). The cellular, D2D and relay users use $C_m$, $D_n$ and $RUE_q$, respectively. The transmitter and receiver of the D2D user are represented by Tn and Rn, respectively. For relay user communication, the RUE-RN link in the band uses the same physical resource blocks (PRBs) as the communication resources of some cellular uplinks in the access subframe. At the same time, to minimize self-interference, RN-BS and RN-BS links use physical resource blocks orthogonal to cellular users in the return subframe.

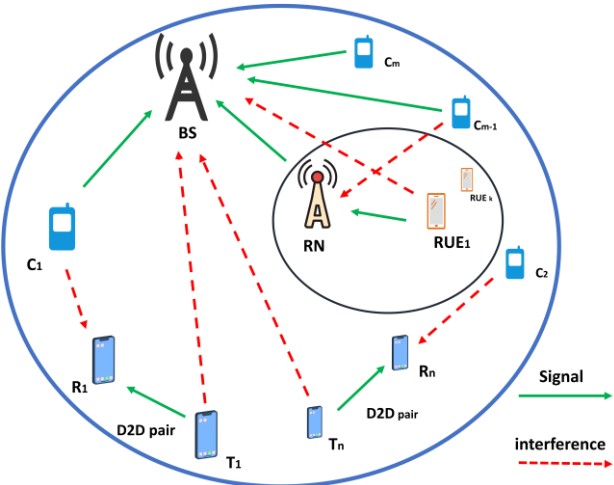

**Figure 1.** D2D communication network system model.

*2.2. Establishment of Problems in System Model*

The signal to interference noise ratio (SINR) of cellular users is:

$$SINR_m = \frac{\alpha_m p_m PL_m}{N_0 + \gamma_n P_n PL_n + \beta_q P_q PL_q} \tag{1}$$

$p_m$ represents the transmission power of the *m*-th cellular user; $p_n$ indicates that the nth D2D uses the transmission power of the kth cell resource fast to the user transmitter; $p_q$ indicates that the *q*-th relay user uses the transmission power of the k-th cell with fast resources; $p_{Lm}$ indicates the link loss between the cellular user and base station; $p_{Ln}$ indicates the interference link loss of the D2D user transmitter to the base station; $p_{Lq}$ indicates the interference link loss of the relay user to the base station; and $N_0$ represents Gaussian white noise, $\alpha_m \in \{0,1\}$, $\beta_q \in \{0,1\}$, $\gamma_n \in \{0,1\}$ where "0" means that the resource block is not reused. "1" means the resource block is reused.

The signal to interference noise ratio (SINR) of the kth RB of a D2D subscriber multiplexed cellular subscriber is:

$$SINR_{n,k} = \frac{\gamma_n P_n PL_n}{N_0 + \alpha_m P_m PL_{m,n} + \beta_q P_q PL_{q,n}} \tag{2}$$

$PL_{m,n}$ represents the interference link loss of the *m*-th cellular user to the *n*-th D2D user receiver; and $PL_{q,n}$ indicates the interference link loss of the *q*-th relay user to the D2D user receiver.

When the relay user multiplexes the *k*-th RB of the cellular user, the signal to interference noise ratio (SINR) of the relay user is:

$$SINR_q = \frac{\beta_q P_q PL_q}{N_0 + \alpha_m P_m PL_{m,q} + \gamma_n P_n PL_{n,q}} \tag{2}$$

$PL_{m,q}$ represents the interference link path loss of cellular users to relay users; and $PL_{n,q}$ indicates the interference link loss of the D2D user transmitter to the relay user.

To sum up, the data transmission rate (total system throughput) of users in the *k*-th resource block in the entire D2D communication system is:

$$R_k = \sum_{M,N,Q} \left\{ log_2 \left( 1 + SINR_m \right) + log_2 \left( 1 + SINR_n \right) + log_2 \left( 1 + SINR_q \right) \right\} \tag{3}$$

This paper studies the joint resource and power allocation problem. Its interface is to maximize the system throughput of all links on each PRB. Its target scalar function can be expressed as:

$$maxR_k = maxR_k \left( \alpha_{m,k}, \beta_{q,k}, \gamma_{n,k}, P_{q,k}, P_{n,k} \right) \tag{4}$$

$$subject \quad to, \quad \sum a_{m,k} \leq 1, \forall m \in M \tag{5}$$

$$\sum \beta_{q,k} \leq 1, \forall q \in Q \tag{6}$$

$$\sum \gamma_{n,k} \leq 1, \forall n \in N \tag{7}$$

$$P_{q,\min} \leq P_{q,k} \leq P_{q,max} \tag{8}$$

$$P_{n,min} \leq P_{n,k} \leq P_{n,max} \tag{9}$$

$P_{q,min}$ and $P_{q,max}$, respectively, represent the minimum transmission power and maximum transmission power allowed by the RUE of the relay communication user equipment; $P_{n,min}$ and $P_{n,max}$, respectively, represent the minimum transmission power and maximum power allowed by the D2D communication transmitter.

## 3. Interference Coordination Algorithm Based on Improved Stackelberg Game

In order to solve the inter layer interference caused by spectrum reuse in D2D communication, this paper proposes an interference coordination algorithm based on the Stackelberg model and the deep reinforcement learning model to adjust the model parameters. Among them, the macro cellular base station (BS) is the leader in the game, the D2D communication user and relay communication user are the followers of the game, the multiplex resource user (D2D/RUE) uses transmission power control and channel resource allocation, and the base station carries out the macro control of the interference cost, thereby adjusting the dynamic balance in the master–slave game model and finally realizing the effective coordination of interference in the D2D communication.

(1) Leader (BS) Utility Function

In the Stackelberg master–slave game model, the utility function of the leader consists of two parts. The first part is its own gain, that is, the throughput of cellular users, and the other part is the channel gain of other users in the system, that is, the channel gain of the D2D and relay users. It is composed of $CUE_m, D2D_n, RUE_q, PRB_k$ and can be expressed as follows:

$$U_{m,n,q,k} = log_2\left(1 + \frac{P_{m,k}PL_m}{N_{0,k} + P_{n,k}PL_n + P_{q,k}PL_q}\right) + \mu_m\left(\gamma_{n,k}P_{n,k}PL_n + \beta_{q,k}P_{q,k}PL_q\right) \tag{11}$$

$$\gamma_{n,k} + \beta_{n,k} = 1 \tag{12}$$

$$\gamma_{n,k} \in \{0,1\} \tag{13}$$

$$\beta_{q,k} \in \{0,1\} \tag{14}$$

$\mu_m$ is the price parameter that each cellular subscriber CUE provides to other links (D2D subscriber multiplex communication link, relay subscriber multiplex communication link) for multiplexing. Formulas (12)–(14) indicate that the value of the multiplexing coefficient of the D2D users and cellular users can only take 1 or 0, that is, only cellular resources can be multiplexed or not multiplexed, and the two cannot take one at the same time, that is, D2D users and relay users cannot reuse the link resources of the same cellular user at the same time.

2. Follower (D2D, RUE) Utility Function

In the Stackelberg master–slave game model, the utility function of the follower consists of two parts. The first part is its own gain, that is, the throughput of D2D users, and the other part is the cost of improving its own gain. The specific utility functions of the D2D and relay users can be expressed as follows:

$$V_{m,n,k} = log_2\left(1 + \frac{P_{n,k}PL_n}{N_{0,k} + P_{m,k}PL_m}\right) - \lambda_m\left(P_{n,k}PL_n\right) \tag{15}$$

$$V_{m,q,k} = log_2 \left( 1 + \frac{P_{q,k} PL_q}{N_{0,k} + P_{m,k} PL_m} \right) - \lambda_m \left( P_{q,k} PL_q \right) \tag{16}$$

### 3.1. Reused Resource User Power Control Algorithm

The transmit power of multiplexed resource users (D2D, RUE) can be obtained by taking the maximum value between the utility function, the potential maximum communication rate gain and the maximum transmit power. Taking the transmission power calculation of D2D users as an example, take the partial derivative function $P_{n,k}$ of the Formula (15) pair, and set the value of the partial derivative to 0 to obtain the best D2D transmission power $P_{n,k}^{m*}$ for the parameters $\lambda_m$.

$$\frac{\partial V_{m,n,k}}{\partial P_{n,k}} = \frac{B}{\ln 2} \frac{PL_n}{N_{0,k} + P_{m,k} PL_m + P_{n,k} PL_n} - \lambda_m PL_n = 0 \tag{17}$$

$$P_{n,k}^{m*} = \frac{B}{\lambda_m PL_q \ln 2} - \frac{N_{0,k} + P_{m,k} PL_m}{PL_n} \tag{18}$$

Similarly, the best D2D transmission power $P_{q,k}^{m*}$, with respect to the parameters, can be obtained by computing the partial derivative function $P_{q,k}$ of the Formula (16) pairs and making the partial derivative value 0.

$$P_{q,k}^{m*} = \frac{B}{\lambda_m PL_q \ln 2} - \frac{N_{0,k} + P_{m,k} PL_m}{PL_n} \tag{19}$$

After obtaining $P_{n,k}^{m*}$ and $P_{q,k}^{m*}$, it is also necessary to limit them to the maximum and minimum transmission power and the transmission power based on the potential maximum communication rate gain.

Assuming that D2D user i multiplexes the channel resources of CU m, the reachability and rate of the D2D user and CU m are expressed as:

$$T\left(P_n^k\right) = log_2 \left( 1 + \frac{p_n^k g_{nn}^k}{P_c g_{kn} + \sigma^2} \right) + log_2 \left( 1 + \frac{P_c g_{mb}^k}{P_n^k g_{nb}^k + \sigma^2} \right) \tag{20}$$

Formula (20) takes the single reuse relationship of D2D users as an example, and its value can reflect the maximum potential of the administration relationship. Therefore, to maximize the transmission power $T\left(P_n^k\right)$ of potential D2D users, the optimization problem can be further described as follows:

$$\max_{p_n^k} T\left(p_n^k\right) \tag{21}$$

$$s.t. \quad 0 \le p_n^k \le p_{max}, \quad \forall n \in N, \quad \forall k \in K$$

In order to ensure the QoS requirements of cellular users after being multiplexed by other users (D2D, RUE), set the SINR threshold of cellular users to $\gamma_c^{th}$. At this time, the transmission power of D2D users can be expressed as:

$$P_{d\,max} = min \left\{ p_{max}, \frac{p_c g_{kb}}{\gamma_c^{th} g_{nb}^k} - \frac{\sigma^2}{g_{nb}^k} \right\} \tag{22}$$

The objective function based on Formula (20) is a concave function about the pair of independent variables. If the stationary point falls in the power range, the power optimal solution can be directly calculated. If the stationary point falls in the right measurement range, the power optimal solution can be obtained. Therefore, the D2D user transmission power $p_{d\,\max}$, based on the potential maximum communication rate gain, can be expressed as:

$$\overline{p}_n^k = \left[ -\frac{\sigma^2}{g_{nb}^k} \pm \sqrt{\frac{g_{nn}^k \sigma^4 - \sigma^2 p_c g_{mb}\left(g_{nb}^k - g_{nn}^k\right) + g_{mb} g_{kn} g_{nb}^k p_c^2}{\sigma^2}} \right]_0^{P_{d\,\max}}$$

To sum up, the transmission power of D2D and relay users can be expressed as:

$$P_{n,k}^{m*} = max\left(min\left(P_{n,k}^{m*}, P_{n,\max}\right), P_{n,\min}, \overline{p}_n^k\right) \tag{23}$$

$$P_{q,k}^{m*} = max\left(min\left(P_{q,k}^{m*}, P_{q,\max}\right), P_{q,\min}, \overline{p}_q^k\right) \tag{24}$$

### 3.2. Resource Allocation Algorithm for Base Station Decision

Taking the sum reported by D2D users $p_{n,k}^{m*}$ and $p_{q,k}^{m*}$ cellular users into Formula 11), the leader's latest utility function matrix $U_{m,n,q,k}$ can be obtained. The size of the matrix is $M \times (N+Q)$. In order to maximize the system throughput and minimize the interference in the D2D communication system while ensuring the communication quality of cellular users, the resource allocation objective function is set as follows:

$$\sum_M \alpha_{m,k} \left( \sum_N \gamma_{n,k} U_{m,n,q,k} + \sum_Q \beta_{q,k} U_{m,n,q,k} \right) \tag{25}$$

As for the calculation of the objective function in Formula (25), the Hungarian resource matching algorithm is used in this paper. The specific steps are as follows:

(1) Traverse all columns in the leader utility function matrix $U_{m,n,q,k}$, select the position $U_{m,n,q,k}$ of the maximum value in the column, and record the row "*m*" corresponding to the maximum value.
(2) Judge whether the values of each "*m*" are different. If the values are the same, execute the fourth step; otherwise, continue to execute.
(3) Find out all the rows "*m*" that do not have duplicate values, record the "n" value of the corresponding D2D user or the "*q*" value of the corresponding relay user, and remove the leader utility matrix (note: the matrix must not be empty because it should be less than), and repeat the first step.
(4) Output all pairs of $(m,n)$ and $(m,q)$ and use the Round Robin algorithm to fairly allocate all resources *k* to $(m,n)$ and $(m,q)$.

### 3.3. DQN-Based Stackelberg Model Interference Coordination Algorithm

In the whole D2D communication interference coordination scheme, the cost parameter $\lambda_m$ plays a crucial role. On the one hand, it determines the transmission power of D2D and relay users, and on the other hand, it can match the best reusable cellular resources for D2D and relay users. In order to achieve the communication quality of effective cellular users when reusing cellular channel resources, this paper proposes an optimization algorithm based on deep reinforcement learning (DQN), which can further

optimize the performance of D2D communication systems by adaptively adjusting the cost parameters $\lambda_m$.

(1) Model Diagram of D2D Communication Interference Coordination Based on DQN

The interference model diagram proposed in this paper mainly consists of the D2D communication environment, transmit power selection of users of multiplexed cellular resources (D2D, CUE), experience pool, deep neural network, Q-Target net, Q-Main estimation network and loss function $L(\theta)$.

This paper will introduce the D2D communication environment, user action selection and reward and punishment functions in the Stackelberg improved algorithm.

(1) Experience Replay Pool

The proposed DQN model is a combination of deep learning and Q learning models. Q learning is an offline learning method (off policy). On the one hand, it can learn the latest strategies according to the user status in the current D2D communication system in real time, and on the other hand, it can learn the past strategies through the experience replay pool. Adding experience replay pool to the whole model greatly improves the learning efficiency of the deep neural network.

The experience replay pool stores multiple transfer learning samples $(s_t, a_t, r_t, s_{t+1})$ obtained by agents interacting with the environment at each time step and stores them in the playback memory network. When the deep learning network is trained, transfer learning samples are added to the experience replay pool, which greatly improves the diversity of the deep learning sample data and ensures cellular users, D2D users and the reliable extraction of link state characteristics of the relay users.

(2) Deep Neural Network CNN Implementation $Q(s, a)$

$Q(s, a)$ refers to the action space from the state of user interference in the current environment of D2D communication to that of the agent based on the current state. In this paper, we use deep learning to fit this mapping efficiently. Specifically, it includes CNN1 (Q-Main Net) and CNN2 (Q-Target Net); the two deep learning network models have the same structure, but their network parameters are completely different. Q-Main Net is used to generate the current Q value; Q-Target Net is used to generate the Target Q value, and this value is used as the reference value of the loss function in depth learning training.

(3) Q-Target Network

The role of the Q-Target network is mainly to disrupt the correlation of system training. In the whole model, Q-Target Net uses the parameters with relatively high service time to estimate the Q value, while Q-Main Net uses the latest parameters to evaluate the latest parameters. As shown in Figure 2, $\max_a Q(s', a'; \theta)$ represents the output of Q-Target Net, and $Q(s, a; \theta_i)$ represents the output of the current network Q-Main Net. The specific process is Q-Target Net and updates the parameters of Q-Main Net according to the loss function. After a certain number of iterations, the network parameters of Q-Main Net are copied to Q-Target Net. After introducing Q-Target Net, the target Q value is kept unchanged during a certain period of training, which reduces the correlation between the current Q value and the target Q value to a certain extent and improves the stability of the algorithm.

(4) Q-Main Network

The Q-Main Net is used to train an infinite approximation function that can approach the real optimal value.

(5) Loss Function: $L(\theta)$

$$L(\theta) = E\left[\left(TargetQ - Q(s,a;\theta)\right)^2\right] \tag{26}$$

Including θ is a network parameter with the following objectives:

$$TargetQ = r + \gamma \max Q\left(s^{'},a^{'};\theta\right) \tag{27}$$

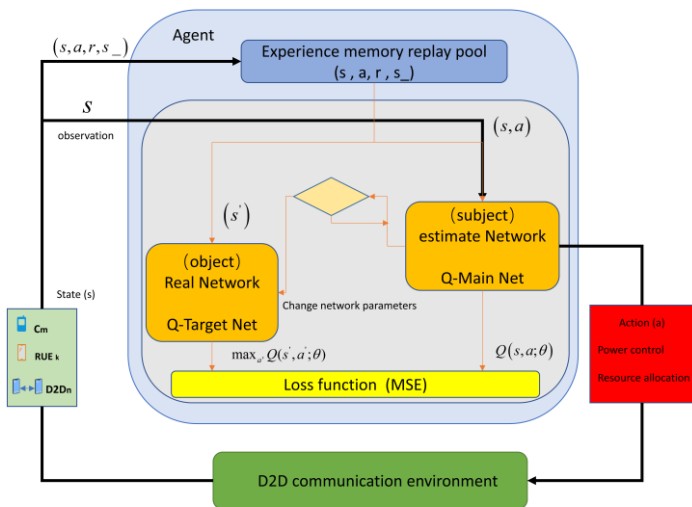

**Figure 2.** Diagram of D2D communication model based on DQN.

(2)   Improved Stackelberg Algorithm

Deep reinforcement learning is often used in the process of the continuous interaction between agents and environmental information. After continuous learning, they perform a series of actions in a specific state to maximize the cumulative return. Compared with deep learning, deep reinforcement learning does not require data sets but only needs to define the agent, state S, action A and reward R.

In this paper, each cellular user can realize the real-time update of triplet variables through continuous interaction and learning with the D2D communication environment information in each time slot t. The variables are state, action and reward. Each component variable of the reinforcement learning model of cost parameters is defined as follows:

Status: defined as the path status of the reusable cellular link on the PRB.

Action: define a group of price factors as actions, and set the value space as:

$$a(t) \in \left(2 \times 10^{15}, 5 \times 10^{21}\right)$$

in this study.

Reward is the reward function: the reward function reflects the learning goal, which is defined as the difference between the logarithmic throughput of the link resources that D2D/RUE reuses CUE and the link resources that D2D/RUE does not reuse CUE, namely:

$$r(s,a) = \begin{cases} R_{m,k} + R_{n,k}, & if\,\gamma_{n,k} = 1, \quad \beta_{q,k} = 0 \\ R_{m,k} + R_{q,k}, & if\,\gamma_{n,k} = 0, \quad \beta_{q,k} = 1 \\ \quad 0 \quad, & otherwise \end{cases} \tag{28}$$

(3)   Updating and Training Algorithm

The above completed the Q table of the mutual mapping of state, action and reward (i.e., return function), state and action in the D2D communication environment based on improved Stackelberg. Two deep learning networks, namely, the Q-Target network and

Q-Main estimation network, were used to fit the Q table. Finally, the algorithm was updated according to the reward value obtained in each iteration of training, and the optimal action selection under different interference states of D2D communication was the output in the experiment. The interference management model based on D2D communication is shown in Figure 3. The detailed algorithm process is explained as follows:

(1)  Experience will play back the storage of the pool

According to the previously determined D2D communication interference environment state parameters, determine the action selection of the agent at this time. There are two kinds of action choices here. One is based on probability $\varepsilon$, which selects an action corresponding to the maximum Q value through the Q-Network, and the other is based on probability $1-\varepsilon$, which randomly selects an action through the action space of the benchmark. (It is worth noting that the value of $\varepsilon$ is set very small from the beginning of training to the time when the experience pool is full, so as to ensure that the previous actions are random and increase the diversity of samples. In addition, the relevant parameters in the Q-Target network are random.) After the action selection, the intelligent agent will execute this action in the D2D communication environment, and then the D2D communication environment will return to the next D2D communication environment state (S_) and reward (R); the new quaternion is (S, A, R, S_), and it is stored in the experience pool. Finally, with the new status (S_) just obtained as the current D2D communication state, repeat the above steps until the experience pool is full.

(2)  Update the parameters of Q-Main Net and Q-Target Net

After the storage of the experience pool is completed, the deep learning model containing the dual deep learning network starts to update. First, the next state (S_) in the D2D communication environment in the experience pool is updated. The reward value (R) after the execution of the action with the current user (D2D user, relay communication user) is brought into the Q-Target network, and the next Q value (y) is calculated. Then, the y value and loss function $L(\theta)$ are used to update the parameters in the Q-Main Net. Then, repeat the above steps. The agent interacts with the D2D communication environment information to generate a new round of (S, A, R, S_). The quads are stored in the experience pool, and the stored samples in the experience pool are used to update the Q-Main Net parameters. With continuous learning, the difference between the two will continue to decrease until the Q-Main Net completes convergence. At this time, the action is the best choice.

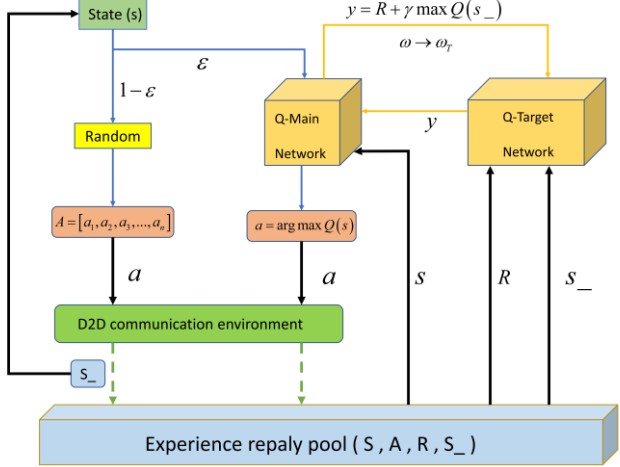

**Figure 3.** D2D communication interference management model based on DQN.

Pseudo code is shown in Algorithm 1:

---

**Algorithm 1:** Pseudo code

---

Initialize replay memory D to capacity N

Initialize action-value function Q with random weights $\theta$

Initialize target action-value function $\tilde{Q}$ with weights $\theta^- = \theta$

For episode = 1, M do

  Initialize sequence $s_1 = \{x_1\}$ and preprocessed sequence $\phi_1 = \phi(s_1)$

  For t=1, T do

    With probability $\varepsilon$ select a random action $a_t$

    Otherwise select $a_t = argmax_a Q(\phi(s_t), a; \theta)$

    Execute action $a_t$ in emulator and observe reward $r_t$ and sequence $x_{t+1}$

    Set $s_{t+1} = s_t, a_t, x_{t+1}$ and preprocess $\phi_{t+t} = \phi(s_{t+1})$

    Store transition $(\phi_t, a_t, r_t, \phi_{t+1})$ in D

    Sample random minibatch of transitions $(\phi_j, a_j, r_j, \phi_{j+1})$ from D

$$set \quad y_i \begin{cases} r_j, & if \quad episode \quad terminates \quad at \quad step \quad j+1 \\ r_j + \gamma \max_a \tilde{Q}(\phi_{j+1}, r_j, \phi_{j+1}), & otherwise \end{cases}$$

    Perform a gradient descent step on $\left(y_j - Q(\phi_j, a_j; \theta)\right)^2$ with respect

    to the network parameters $\theta$

    Every C step reset $\tilde{Q} = Q$

  End For

End For

---

## 4. Simulation Experiment Results and Analysis

This paper considers a 500 m radius cellular D2D communication network, in which there are 30 cellular users (CUE, RUE), and the relay coverage is 0.35 × 500 m. The specific parameter settings are shown in Table 2.

**Table 2.** System simulation parameters.

| Simulation Parameters | Parameter Value (Unit) |
|---|---|
| Frequency | 28 GHz |
| System bandwidth | 1 GHz |
| Cell radius | 500 m |
| Number of cellular users (CUE, RUE) | 30 |
| Coverage of relay nodes | 0.35 × 500 m |
| Road loss model of Cellular link | 3GPP TR36.814 V9 A.2.1.1.2 |
| Road loss model relay link | 3GPP TS 36.814 V9.0.0 |
| Road loss model of D2D link | 3GPP TR36.814 V12 A.2.1.2 |
| Noise power spectral density | −174 dBm/Hz |
| Maximum transmit power of D2D link | 24 dBm |
| D2D communication link interference threshold | −105 dBm |

This paper selects two benchmark D2D communication interference management algorithms for algorithm comparison. The first is a D2D communication interference coordination algorithm (RM) based on random multiplexing, in which D2D and relay users randomly multiplex cellular users. The second is based on Stackelberg's D2D communication interference management algorithm (ST), which uses game theory to multiplex D2D and relay users to cellular users. This paper proposes a D2D communication interference coordination scheme based on a DQN-improved Stackelberg game. The scheme uses the idea of mutual game between relay and D2D users to dynamically and adaptively obtain the reuse of cellular user resource blocks by users in D2D communication networks. In order to ensure that the simulation experiment has certain reference value, this paper selects average throughput and generalized proportional fairness (GPF), etc.

GPF is defined as follows:

$$GPF = \frac{1}{|\Gamma|}\sum_{x \in \Gamma} log R_x \tag{29}$$

Figure 4 shows the cumulative distribution function of cellular user throughput under different interference coordination algorithms. It can be seen from the figure that, compared with the RM and ST algorithms, the algorithm proposed in this paper has stronger anti-interference ability in the 0–1000 kbps interval. This is because the optimization effect of the RM algorithm is random. The ST algorithm uses the interference strategy of the master–slave game that relies too much on D2D users. It can be seen from Figure 4 above that no matter which algorithm is used, the cellular cumulative distribution function will tend to be stable and gentle at the end of the experiment. This is because the interference in the system is complex and changeable at the beginning of the system, but as the interference coordination algorithm starts to work over time, the anti-interference ability of the system is enhanced and finally tends to be a dynamic balance.

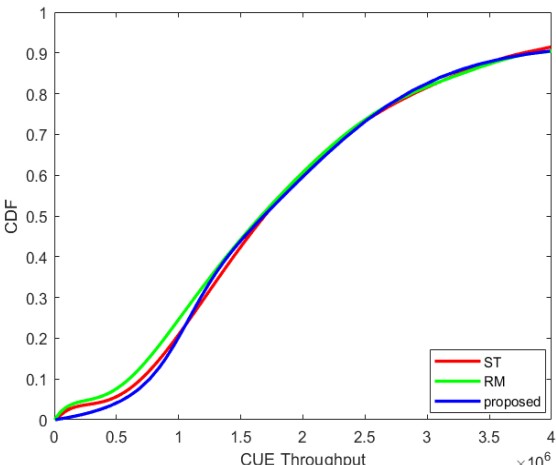

**Figure 4.** CUE throughput CDF.

Figure 5 shows the GPF of cellular users with different D2D users under the optimization of three different interference coordination algorithms. It can be seen from the figure that, compared with the other two algorithms, the performance of the scheme proposed in this paper is always better than that of the RM algorithm because the idea of the dynamic game is not introduced into the D2D communication interference coordination. Compared with the ST algorithm, when there are 3–5 D2D users, the optimization effect of the algorithm proposed in this paper is slightly lower than that of the ST algorithm, and when there are 6–9 D2D users, the optimization effect is better than that of the ST algorithm. This is because at the beginning of repeated games, both sides of the game have

unique selfish attributes, and then as time goes by, both sides share dynamic resource blocks interactively.

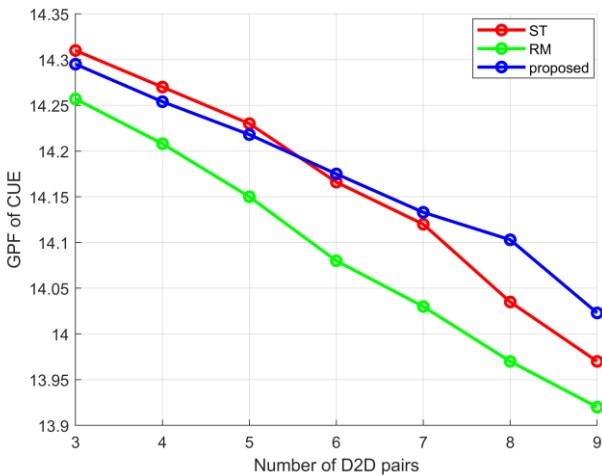

**Figure 5.** GPF of CUE under different D2D logarithms.

Figure 6 shows the broken line chart of the 5% minimum throughput change in cellular users under three different interference coordination algorithms and different D2D users. It can be seen from the figure that, compared with the other two algorithms, when the number of D2D users increases from three to six and the RM algorithm is used, the throughput of 5% of the lowest throughput of the cellular user decreases from 470 to 190 kbps; when the ST algorithm is used, its throughput decreases from 640 to 220 kbps. The algorithm proposed in this paper is reduced from 700 to 380 kbs, which has certain advantages in the interference coordination with the RM and ST algorithms. On the one hand, it proves that the interference of iterative deep reinforcement learning and game theory can be used together. On the other hand, it proves the correctness of the algorithm improvement in this paper.

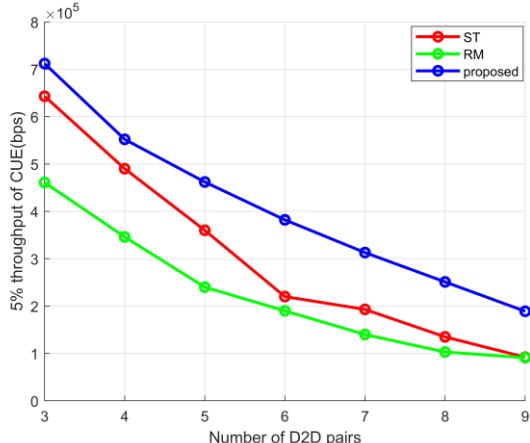

**Figure 6.** The 5% minimum throughput of CUE under different D2D logarithms.

Figure 7 shows the dotted line changes of cellular users' GPF at different distances between the relay node and the base station under the optimization of three different interference coordination algorithms. It can be seen from the figure that with the increase in distance, the three GPF dotted lines show an upward trend. This is because the distance between the edge users and relay nodes in the cellular networks is getting closer and closer, and the communication quality of the edge users is effectively improved. At the

same time, compared with the other two algorithms, the GPF of the cellular base station is always the highest after using the algorithm in this paper, and the curve variation trend of the algorithm in this paper is relatively small, which once again proves that the algorithm has good stability.

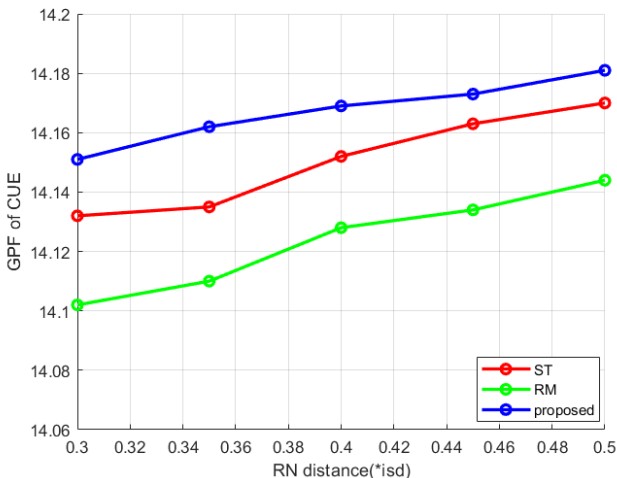

**Figure 7.** GPF of CUE at different distances (REU-BS).

Figure 8 shows the dotted line change in the 5% minimum throughput of cells at different distances between the relay nodes and base stations under the optimization of three different interference coordination algorithms. It can be seen from the figure that with the increase in the distance between the relay nodes, the minimum throughput of cellular users increases by 5% after using the three algorithms. The minimum throughput of 5% of the cellular users in the RM algorithm optimization strategy shows a relatively large change, while the change in the broken line after using the other two algorithms is relatively stable. The algorithm proposed in this paper is the best. This is because the ST algorithm and the algorithm proposed in this paper both adopt the idea of a dynamic game, and the algorithm in this paper uses the autonomous game more deeply than the RM algorithm and uses the depth to strengthen the environmental interaction in learning, which increases the algorithm's ability to mine the interference state in the network and greatly improves the system's interference management ability.

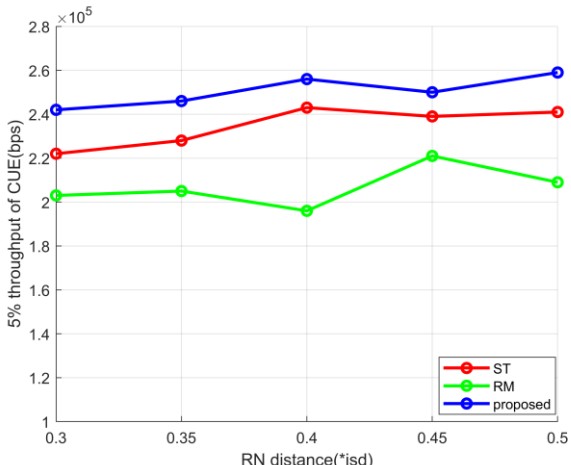

**Figure 8.** The 5% minimum throughput of CUE at different distances (REU-BS).

## 5. Conclusions

In this paper, we propose a D2D communication interference management scheme based on the improved Stackelberg model and apply it to the D2D heterogeneous cellular network where D2D, relay and cellular users coexist. The price parameters of the Stackelberg model are dynamically improved by using the idea of deep reinforcement learning, which realizes the dynamic adjustment of channel allocation and power control in the network, improves the communication quality of users and greatly improves the interference management capability of the system.

With the rapid development of the communication network, the network will be more heterogeneous and diversified in the future, and the network interference in the system will be more complex. The next step will introduce the idea of confrontation networks into the existing in-depth reinforcement learning to further optimize the impact of the cost function in the Stackelberg model on the multi heterogeneous D2D communication interference and improve the interference management capability of the system.

## 6. Patents

A patent entitled "Heterogeneous cognitive wireless sensor network cluster routing method" is disclosed under CN110708735B.

**Author Contributions:** Conceptualization, X.L. and G.C. (Guangjiao Chen); methodology, X.L.; validation, X.L., G.C. (Guifen Chen), and G.C. (Guangjiao Chen); data curation, X.L.; writing—original draft preparation, X.L.; writing—review and editing, X.L.; visualization, X.L.; project administration, G.C. (Guifen Chen), G.W., and Z.S.; funding acquisition, G.C. (Guifen Chen). All authors have read and agreed to the published version of the manuscript.

**Funding:** This research was funded by the "Thirteenth Five-Year Plan" Science and Technology Research Project of Jilin Provincial Department of Education, Research on Large-scale D2D Access and Traffic Balancing Technology for Heterogeneous Wireless Networks JJKH20181130KJ, Special Project on Industrial Technology Research and Development of Jilin Province, Research on Self-organizing Network System of Unmanned Platform for Optoelectronic Composite Communication, 2022C047-8.

**Institutional Review Board Statement:** This study does not involve humans or animals.

**Informed Consent Statement:** Not applicable.

**Data Availability Statement:** Not applicable.

**Conflicts of Interest:** The authors declare no conflict of interest.

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
