# Peer review of "D2D Communication Network Interference Coordination Scheme Based on Improved Stackelberg"

_sustainability, doi:10.3390/su15020961_

Round 1

Reviewer 1 Report

The technical details look good to me. Here are my comments:

Line 21. Qos => QoS

Line 29. Seems that [1-2] is in another font.

Lines 138-144. I suggest using another enumeration format. In my first sight, I think that (1), (2) and (3) are referring to some equations. It may be better if a new item starts in a new line.

Lines 151-152. I think a formal way is to write something like j = 1, 2, 3, ..., M. I mean, there should have a pair of commas before and after ...  On the other hand, it seems that these math expressions are not in math mode (or math font style).

Line 152. Should the m, n and q in Cm, Dn, and RUEq be subscripts? (i.e., like the m of P_m in line 161)

Line 161. Seems that "Where" should not be in a new paragraph and should not be capitalized.

Line 163. I can see qth and k-th in the same line. Please be consistent, say, either qth and kth, or q-th and k-th.

Lines 161-166. Almost all words right after each math notation are capitalized. They should be started with small letter because they are in the middle of a sentence.

Line 167. "Where" should not be capitalized.

Line 255. s.t => s.t.

Line 266. The Pd max in the superscript at the end of the equation is not in its correct form. In previous lines, it should be p_{d max}. That is, the d max should be in the subscript of p, and the p was a small letter.

Lines 271 and 408. I suggest moving the (sub)section title to the next page. It is a bit strange to have a section title at the bottom of a page.

Line 347. Should TatgetQ be TargetQ?

Line 349. The typesetting of TargetQ is screwed. Currently it is T arg etQ.

Figure 6. The caption is now in the next page. By the way, what is the 5ile in the y-axis title?

Author Response

 Dear Reviewers:

Thank you very much for your valuable comments and careful guidance. We have carefully revised the full text and marked it in red (The coordinate axis parameters in Figure 6 have been changed. Thank you for your guidance). Thank you again and wish you a happy life.

 Special thanks to you for your good comments.

                                                                                      Yours sincerely,

                                                                                                   Chen

Reviewer 2 Report

Aiming at the problem that the existing D2D communication net-12 work system has complex user interference and the communication quality of cellular users is dif-13 ficult to guarantee, a D2D communication network interference coordination scheme based on im-14 proved stackelberg has been proposed in the paper. The authors need to futher improve the following problems. 

In the experimental part, the author needs to show the advantages of the proposed method with specific comparative analysis.

2 Just suggest authors carefully improve English presentation of this paper.

Author Response

Dear Reviewers:

Thank you for your letter and for the reviewers’ comments concerning our manuscript entitle” D2D Communication Network Interference Coordination Scheme Based on Improved Stackelberg” (ID: sustainability-2121626). Those comments are all valuable and every helpful for revising and improving our paper, as well as the important guiding significance to our researches. We have studied comments carefully and have made correction which we hope meet with approval. Revised portion are marked in red in the paper. The main correction in the paper and the responds to the reviewer’s comments are as following:

Response question 1:

Thank you for your good comments. In the simulation experiment stage, this paper selects two benchmark D2D interference management algorithms for comparison. One is the D2D communication interference management algorithm based on random multiplexing, which is universal. The second one selects the stackelberg based D2D communication interference management algorithm related to the algorithm proposed in this paper. This algorithm can be compared with the algorithm proposed in this paper in terms of specific advantages.

Then, this paper discusses the cumulative distribution function of the cellular user throughput, the GPF of the cellular user when the number of D2D users is increasing, the 5% minimum throughput of the cellular user when the number of D2D users is increasing, and the GPF and 5% minimum throughput of the cellular user when the distance between the relay node and the base station is increasing. These aspects prove that the algorithm proposed in this paper has certain advantages in both universality and particularity.

For the problem that you have given you to show more advantages of the algorithm in this paper, first of all, we carefully conceived and summarized the experimental simulation part, and made changes. Because of the urgency of submission time and the long time required for in-depth intensive learning experiment training, we will add additional experiments in the future. Please forgive us for the inconvenience caused.

Response question 2:

For the English expression of this article, we have carefully revised the full text to improve the quality of scientific research and English expression.

Special thanks to you for your good comments.

                                                                                       Yours sincerely,

                                                                                                    Chen

Reviewer 3 Report

The paper is well written and well structured. Though the paper has lot of simulation methods used, it failed to explain how it overcame the cons of those specific algorithms used.

Spell out each acronym the first time used in the Abstract as well as the body of the paper.

Gap in the article is not presented properly by the authors

Authors failed to highlight contributions in this article

There is no section about related works. Use a table which able to differentiate between existing works and identify the limitations.

Please check the legends of figure 6 and 8 (5ile throughput of CUE(bps))

Please cite each equation and clearly explain its terms.

Clearly highlight the terms used in the algorithm and explain them in the text.

In conclusion section, there must be maximum two paragraphs. The first paragraph is for briefly discussing the entire paper and the second paragraph is for discussing some future works.

Author Response

Dear Reviewers:

Thank you very much for your valuable comments and careful guidance. We have carefully revised the full text and marked it in red.

Response question1:

The acronyms of the abstract and the full text have been changed, and the blank space in the article has been changed in the latest manuscript. Please check.

Response question2:

For the table of outstanding contributions and related work in this article, we have added it in the introduction. Please check it.

Response question3:

As for the legend of Figures 6 and 8, we have made changes in the latest manuscript. Thank you for your guidance.

We apologize for the explanation of terms in mathematical equations and algorithms. The latest manuscript has completed the explanation of algorithm terms and the change of equation terms.

Response question4:

For the conclusion part, we have completed two paragraphs of this part, please check.

 Special thanks to you for your good comments.

                                                                                        Yours sincerely,

                                                                                                     Chen

Reviewer 4 Report

minor English revisions required

Author Response

 Dear Reviewers:

Thank you very much for your valuable comments and careful guidance. We have carefully revised the full text to improve the quality of scientific research and English expression.

 Special thanks to you for your good comments.

                                                                                    Yours sincerely,

                                                                                           Chen

Round 2

Reviewer 3 Report

Paper can be accepted in its current form.